# Defining the determinants of protection against SARS-CoV-2 infection and viral control in a dose-down Ad26.CoV2.S vaccine study in nonhuman primates

Daniel Y. Zhu[1], Matthew J. Gorman[2], Dansu Yuan [2], Jingyou Yu [3], Noe B. Mercado[3], Katherine McMahan[3], Erica N. Borducchi[3], Michelle Lifton[3], Jinyan Liu[3], Felix Nampanya[3], Shivani Patel[3], Lauren Peter[3], Lisa H. Tostanoski [3], Laurent Pessaint[4], Alex Van Ry[4], Brad Finneyfrock[4], Jason Velasco[4], Elyse Teow[4], Renita Brown[4], Anthony Cook[4], Hanne Andersen[4], Mark G. Lewis [4], Douglas A. Lauffenburger[1], Dan H. Barouch[2,3,5], Galit Alter [2]*

1 Department of Biological Engineering, Massachusetts Institute of Technology, Cambridge, Massachusetts, United States of America, 2 Ragon Institute of MGH, MIT and Harvard, Cambridge, Massachusetts, United States of America, 3 Center for Virology and Vaccine Research, Beth Israel Deaconess Medical Center, Harvard Medical School, Boston, Massachusetts, United States of America, 4 Bioqual, Rockville, Maryland, United States of America, 5 Massachusetts Consortium on Pathogen Readiness, Boston, Massachusetts, United States of America

☯ These authors contributed equally to this work.
* galter@mgh.harvard.edu

## Abstract

Despite the rapid creation of Severe Acute Respiratory Syndrome Coronavirus 2 (SARS-CoV-2) vaccines, the precise correlates of immunity against severe Coronavirus Disease 2019 (COVID-19) are still unknown. Neutralizing antibodies represent a robust surrogate of protection in early Phase III studies, but vaccines provide protection prior to the evolution of neutralization, vaccines provide protection against variants that evade neutralization, and vaccines continue to provide protection against disease severity in the setting of waning neutralizing titers. Thus, in this study, using an Ad26.CoV2.S dose-down approach in non-human primates (NHPs), the role of neutralization, Fc effector function, and T-cell immunity were collectively probed against infection as well as against viral control. While dosing-down minimally impacted neutralizing and binding antibody titers, Fc receptor binding and functional antibody levels were induced in a highly dose-dependent manner. Neutralizing antibody and Fc receptor binding titers, but minimally T cells, were linked to the prevention of transmission. Conversely, Fc receptor binding/function and T cells were linked to antiviral control, with a minimal role for neutralization. These data point to dichotomous roles of neutralization and T-cell function in protection against transmission and disease severity and a continuous role for Fc effector function as a correlate of immunity key to halting and controlling SARS-CoV-2 and emerging variants.

**Data Availability Statement:** Data can be found in the following Github repository: https://github.com/dzhu8/Ad26-Dose-Down.

**Funding:** We (D.L. and D.Y.Z.) acknowledge support from National Institutes of Health Grant U19-Ai42790. The funders had no role in study design, data collection and analysis, decision to publish, or preparation of the manuscript.

**Competing interests:** I have read the journal's policy and the authors of this manuscript have the following competing interests: GA is the founder of SeromYx Systems Inc. D.H.B. is a co-inventor on provisional vaccine patent (63/121,482; 63/133,969; 63/135,182).

**Abbreviations:** ADCD, antibody-dependent complement deposition; ADCP, antibody-dependent cellular phagocytosis; ADNKA, antibody-mediated NK cell activating; ADNP, antibody-dependent neutrophil phagocytosis; BAL, bronchoalveolar lavage; COVID-19, Coronavirus Disease 2019; FcR, Fcγ receptor; IACUC, Institutional Animal Care and Use Committee; IQR, interquartile range; IT, intratracheal; LASSO, least absolute shrinkage and selection operator; NHP, nonhuman primate; NS, nasopharyngeal swab; NTD, N-terminal domain; PCA, principal component analysis; PLS-PM, partial least squares path modeling; PLS-R, partial least square regression; RBD, receptor-binding domain; RLU, relative light unit; RPMI, Roswell Park Memorial Institute; SARS-CoV-2, Severe Acute Respiratory Syndrome Coronavirus 2; sgRNA, subgenomic RNA; VOC, variant of concern; VIP, variable importance; VP, viral particle; WT, wild-type.

## Introduction

Since December 2019, Severe Acute Respiratory Syndrome Coronavirus 2 (SARS-CoV-2) has spread globally, linked to greater than 200 million confirmed cases and over 4 million deaths [1]. Several vaccines have been granted emergency use authorization globally [2–7], affording variable levels of efficacy, particularly against emerging viral variants of concern (VOCs) [8–13]. While neutralizing antibodies represented an early robust surrogate of immunity prior to the evolution of VOCs, vaccine breakthroughs are rapidly emerging globally in the absence of concurrent levels of disease [10–12]. Moreover, vaccines afford protection against severe disease and death prior to the evolution of neutralization [14–16] as well as in the setting of waning immunity [17], suggesting that alternate immunological mechanisms may exist, which provide protection against disease, including antibody-mediated Fc effector functions [18–21] and T cells [22–26], which appear to be less affected by VOCs [9,27]. However, how neutralization, Fc effector function, and T cells collectively contribute to durable and effective protection against SARS-CoV-2 and emerging variants remains incompletely understood.

To begin to dissect the protective role of each immune mechanism, we exploited a vaccine dose-down strategy, aimed at inducing progressively lower SARS-CoV-2 immunity, to capture the immunological breaking point of immune protection upon challenge. Neutralizing antibodies, T-cell responses, and Fc effector functions [28] were captured at peak immunogenicity across all dosing groups prior to challenge. While antibody titers and neutralization were minimally affected by dosing, Fc profiles were highly dose dependent. Similarly, recognition of VOCs was also highly dose dependent. Protection against infection was robustly linked to neutralizing antibody titers and Fc effector mechanisms. Conversely, viral control following infection was associated with Fc effector functions and T cells. Thus, Fc effector function clearly synergizes with both neutralization and T cells, to block transmission or control viremia, respectively. These data point to disparate functions of neutralizing antibodies and T cells in antiviral immunity, but a consistent role of Fc effector functions across both transmission and protection against disease.

## Results

### Dose-dependent upper respiratory SARS-CoV-2 replication following challenge

To more deeply understand the correlates of protection, 5 independent groups of rhesus macaques ($n = 5$) were immunized with different doses of the Ad26.CoV2.S vaccine ($1 \times 10^{11}$, $5 \times 10^{10}$, $1.125 \times 10^{10}$, and $2 \times 10^9$ viral particles [VPs]) or a sham control (**Fig 1A**). On day 42, all macaques were challenged with $1.0 \times 10^5$ TCID50 ($1.2 \times 10^8$ RNA copies) of SARS-CoV-2, and viral replication was assessed by analyzing subgenomic RNA (sgRNA) levels in the upper and lower respiratory tracts by way of nasopharyngeal swabs (NSs) and bronchoalveolar lavage (BAL), respectively, on days 1, 2, 4, 7, and 10 postchallenge. As previously published in [29], vaccine doses as low as $2 \times 10^9$ VPs provided robust protection in the lower respiratory tract, whereas protection in the upper respiratory tract showed a negative dose dependency (**Fig 1B**), in which a higher dose ($1.125 \times 10^{10}$ VPs) was required for protection. Additionally, as previously reported [30], higher doses generated consistently higher binding antibody titers and neutralizing titers, in addition to stimulating increased T-cell cytokine production above an observed threshold (**Fig 1B**) in response to both SARS-CoV-2 spike and receptor-binding domain (RBD). This illustrates the complete protection provided by high doses of the vaccine that wanes as a function of dose and highlights the need to deeply mine for correlates of protection in the upper respiratory tract.

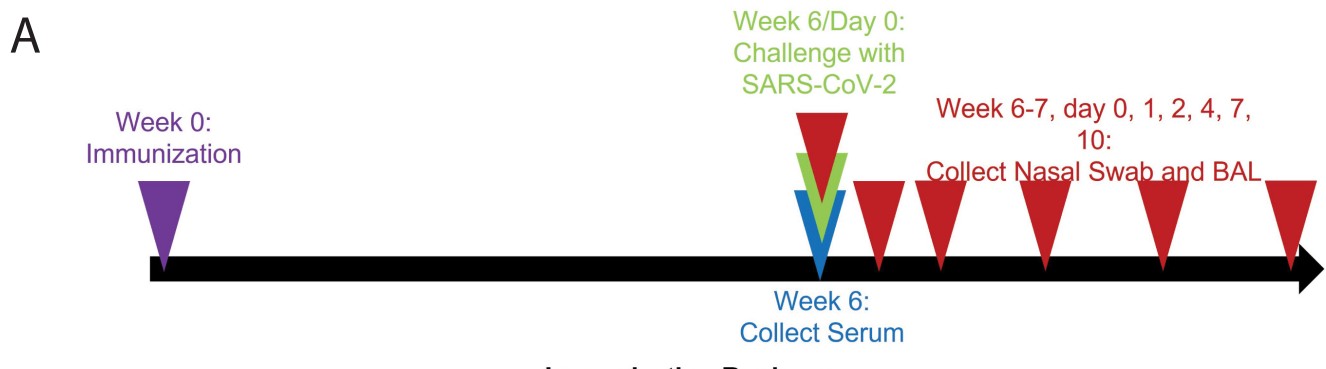

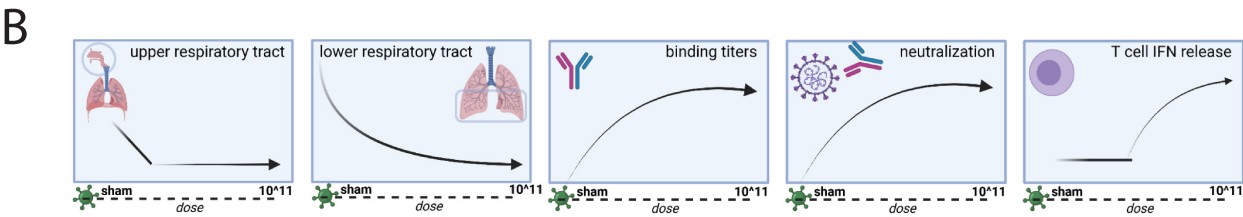

**Fig 1. AD.COV2.S dose-down immunization scheme and graphical depiction of the immune response. (A)** Rhesus macaques were immunized with 5 different doses of Ad26.COV2.S, profiled for their immune response, and challenged with SARS-CoV-2. The timeline shows the timing of immunization (week 0), serum collection (week 2, 4, and 6), SARS-CoV-2 challenge (day 43), and viral titer sampling of the cohort (day 0, 1, 2, 4, 7, and 10 postinfection). The legend states the dose of Ad26.COV2.S given to each group of NHP. **(B)** From left to right, summary of the observed trends of viral sgRNA in the upper respiratory tract, sgRNA in the lower lower respiratory tract, binding antibody titer, neutralizing antibody titer, and T-cell IFNγ secretion as they relate to dose. Fig 1B was created with BioRender.com. BAL, bronchoalveolar lavage; IFNγ, interferon gamma; NHP, nonhuman primate; SARS-CoV-2, Severe Acute Respiratory Syndrome Coronavirus 2; sgRNA, subgenomic RNA; VP, viral particle.

### Dosing effects on cellular and humoral immune responses

To further dissect the qualitative effects of vaccine dosing on the functional humoral immune response, systems serology was applied to the aforementioned BAL and NS samples, collected on day 43 prior to SARS-CoV-2 challenge. Similar to the neutralizing antibody titers, the 2 highest doses (I and II) induced similar levels of anti–SARS-CoV-2 IgG1, IgG2, IgG3, IgG4, and IgA antibody titers, with slightly lower levels at dose III and significantly lower levels at dose IV (**Fig 2A–2F**). Conversely, while the 2 highest doses also generated equivalent levels of Fcγ receptor (FcR) binding titers (**Fig 2G and 2H**), more significant loss of FcR binding was observed in dose group IV, with a broad spread of binding to the FcRs. These data point to a more significant impact of dosing on shaping the overall magnitude of the FcR binding compared to driving titers, neutralizing antibody responses or T cells.

To further explore the impact of dosing on Fc effector function, we finally profiled the vaccine-induced Spike-specific response. Similar to FcR binding profiles, similar antibody effector functions were observed in the top 2 dose groups (**Fig 2I–2L**), with a significant decline in antibody-dependent cellular phagocytosis (ADCP) and antibody-dependent neutrophil phagocytosis (ADNP) in group III and significant decline in ADCP, ADNP, antibody-dependent complement deposition (ADCD), and antibody-mediated NK cell activating (ADNKA) activity in group IV. Together, these findings suggest that antibody FcR binding and Fc effector function are more sensitive to changes in vaccine dosing, compared to traditional metrics

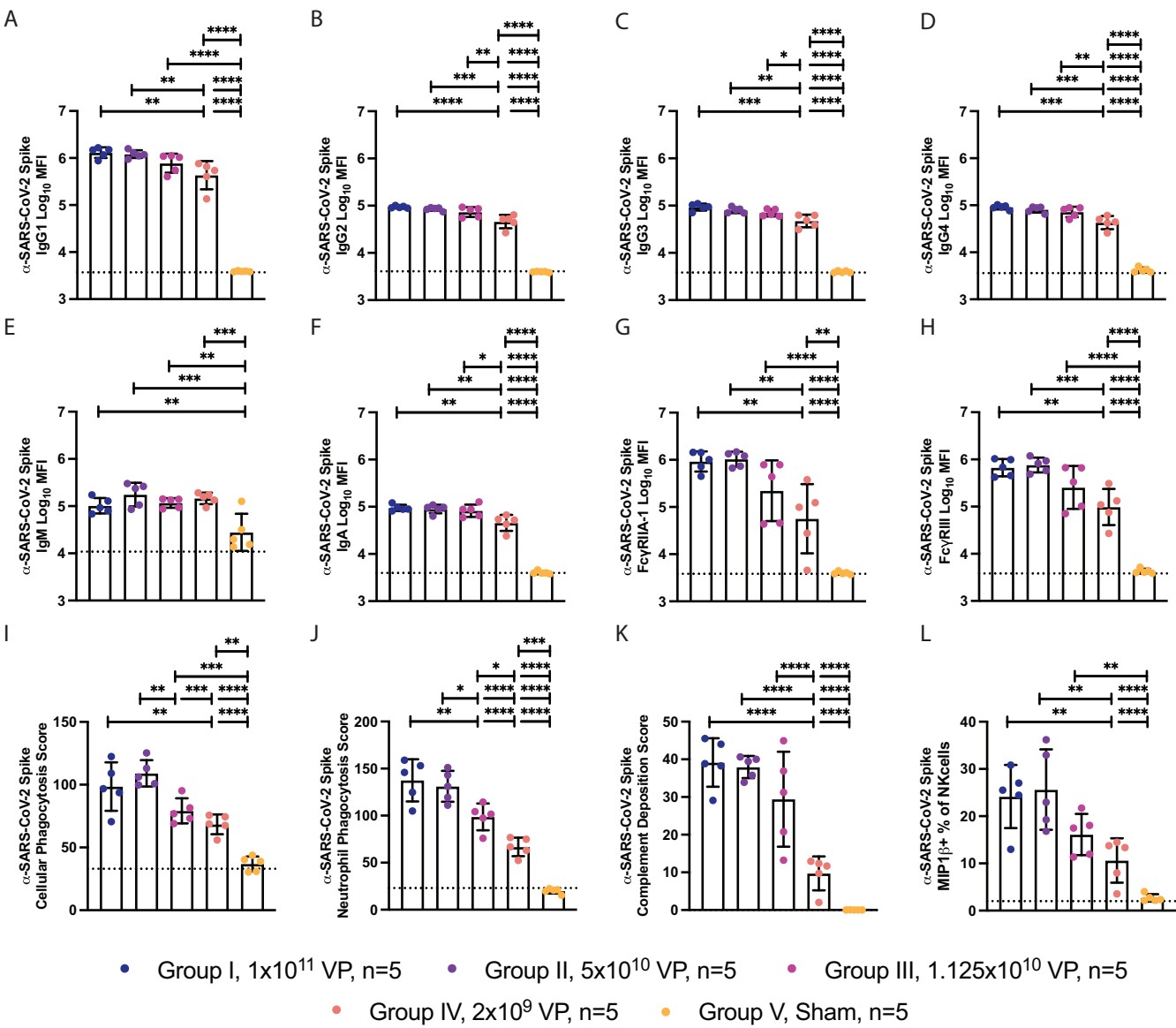

**Fig 2. System serology profiling of anti-Spike antibodies from Ad26.COV2.S vaccinated NHPs. (A)** The humoral response against SARS-CoV-2 Spike was profiled using system serology. **(B–G)** The titer of IgG1 (B), IgG2 (C), IgG3 (D), IgG4 (E), IgM (F), and IgA (G) antibodies against SARS-CoV-2 were profiled using Luminex. **(H, I)** The titer of anti–SARS-COV-2 antibody binding to FcγRIIA-1 (H) and FcγRIII (I) were profiled using Luminex. **(J–M)** The graphs represent the ability of the humoral response to stimulate ADCP (J), neutrophil phagocytosis (ADNP) (K), complement deposition (ADCD) (L), or NK cell activation (NKdegran) (M) when stimulated with antigen-coated beads or plates. Bars represent the mean, and error bars represent the standard of deviation. This figure can be generated from the data found in data/figure2_data.csv of https://github.com/dzhu8/Ad26-Dose-Down. ADCD, antibody-dependent complement deposition; ADCP, antibody-dependent cellular phagocytosis; ADNP, antibody-dependent neutrophil phagocytosis; MFI, median fluorescent intensity; NHP, nonhuman primate; NK, natural killer; SARS-CoV-2, Severe Acute Respiratory Syndrome Coronavirus 2; VP, viral particle.

of immunogenicity (neutralizing antibodies, T cells, and titers) and may provide additional insights into correlates of immunity.

## Distinct humoral profiles across vaccine dose regimens revealed by multivariate analysis

Given the different univariate variation across multiple SARS-CoV-2–specific immune responses, we next aimed to define the impact of dose on overall vaccine-induced immune

profiles. To focus on the effect of vaccine-induced immunity, we removed all Sham samples. Striking differences were noted in the overall Spike-specific humoral immune profiles across the 4 dose groups (Fig 3A), marked by the most robust, fully immunoglobin class-switched (no IgM) titers in the animals that received the highest vaccine dose, followed by robust titers, FcR binding, T-cell responses, and Fc effector function in group II immunized animals. Conversely, decreased antibody titers and Fc effector functions were noted in group III immunized animals, with a very significant loss of FcR binding and limited T-cell immune responses. Finally, the lowest antibody titers, Fc effector functions, T-cell responses, and negligible FcR binding were noted in group IV immunized animals. Moreover, additional analysis of the overall architecture of the humoral immune response using Pearson correlations between all humoral variables highlighted the relative conservation of the humoral immune architecture across the vaccine groups (S1 Fig), with the exception of ADCP, which was least dose dependent.

To further define the impact of vaccine dosing on the epitope-specific response across the Spike protein, we also examined the vaccine-induced humoral immune response to the RBD, S1, S2 and N-terminal domain (NTD) (Fig 3B). While heterogenous responses were seen for animals in the same dose group following vaccination, generally, the highest and broadest immune responses were noted in group 1 ($1 \times 10^{11}$ VP) group, with a dose-dependent decrease in titer, FcR binding, and Fc-induced immune function across the other vaccine dose groups for all regions of the SARS-CoV-2 Spike. IgM titers were the exception, with IgM responses anticorrelated with other subclasses, as a marker of mature class switching to IgA and IgG [31]. To finally define whether antibody profiles could distinguish the dose groups, a principal component analysis (PCA) was generated across the 4 vaccine groups (Fig 3C). The 2 top dose groups clustered together, whereas the 2 bottom groups clustered to the left of the PCA plot. The majority of the variance (60%) was explained by the first principal component, and a comparison of the loadings (the coefficients of the linear combination of variables used to construct principal components) along this axis (Fig 3D) revealed enrichment of S2 and Spike FcR binding features as the predominantly enhanced features in animals that received the highest vaccine doses. These data revealed the significant enrichment of FcR binding antibodies among animals that received the higher vaccine doses.

## Neutralizing antibody titers and FcR binding are enriched in protected animals

To begin to define the specific immunologic markers associated with protection against viral challenge, animals were divided into 3 groups, consisting of nonhuman primates (NHPs) exhibiting total protection (no detectable viral replication), partial protection (no detectable viral replication in BAL), and no protection (viral replication in BAL and nasal swab). Polar plots revealed differences in the overall vaccine-induced immune profiles across the groups, with the strongest responses, across immunologic readouts, in animals with total protection (Fig 4A). Intermediate responses were observed in the partially protected group, and lower responses, marked by near complete loss of FcR binding in the animals that were not protected. Unsupervised multivariate analysis (PCA, as above) highlighted a clear separation in the immune profiles of animals that exhibited complete protection against SARS-CoV-2 and animals exhibiting viral load in at least one compartment (Fig 4B). To precisely define the features that tracked with protection, fold changes were computed for each immunologic marker and visualized in a volcano plot (Fig 4C), across animal groups. The most significant fold change, depicted on the y-axis, was neutralizing antibody titers (as assessed using the nonparametric Mann–Whitney U test), with all humoral features save for ACDP, NK-secreted CD107a, and IgM titer also found to be significant after multiple hypothesis correction using

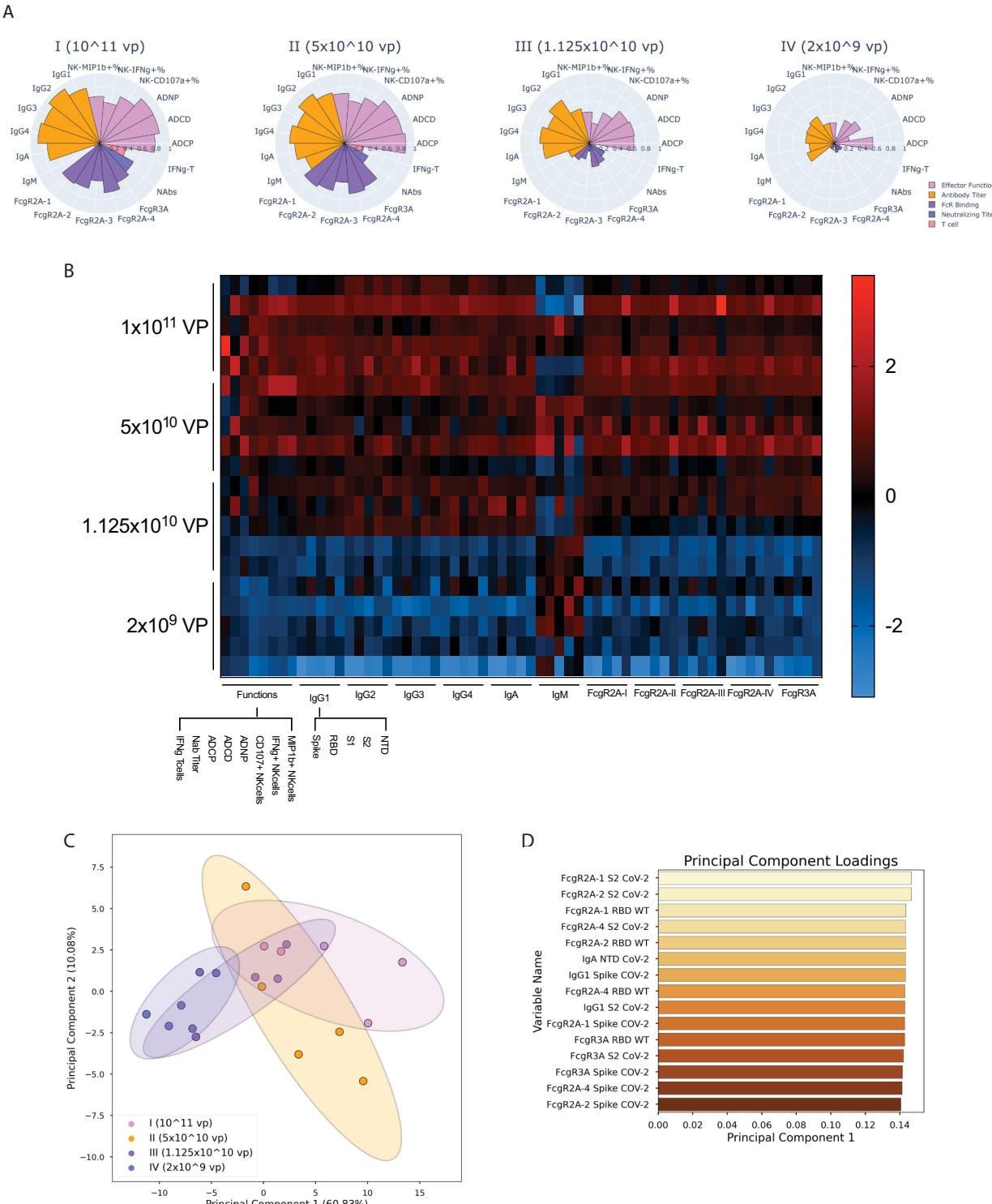

**Fig 3. Multivariate analysis of the humoral response among each vaccine regimen.** Further analysis was performed to identify which humoral features defined the response in each dose. **(A)** Polar plots showing the mean percentile of each of the anti–SARS-CoV-2 antibody titers, FcR binding, and effector function activation features for NHPs, grouped by the approximate number of VPs received in the vaccine dose. Each wedge represents an individual feature, with colors representing the type of feature: orange: effector function; lavender: antibody titer; violet: FcR binding; indigo: neutralizing titer; and pink: T-cell cytokine secretion. **(B)** The heatmap shows the humoral response, antibody isotype, and FcR binding

titers, to multiple SARS-CoV-2 proteins (Spike, S1, S2, and RBD), and NTD, as well as the functional responses to Spike. The squares represents the z-scored value for each immune feature (x-axis) for each NHP (y-axis). Z-scored was calculated across each individual column. **(C)** A latent variable scores plot resulting from PCA shows the degree of separation between the 4 vaccine doses that was achieved by analyzing anti–SARS-CoV-2 (anti-S1, S2, NTD, RBD, and Spike) features. Each point is an individual NHP, color coded by vaccine dose group (purple: $10^{11}$ VPs, orange: $5 \times 10^{10}$ VPs, blue: $1.125 \times 10^{10}$ VPs, and pink: $2 \times 10^{9}$ VPs). Ellipses illustrate the 95% confidence interval for dose groups. **(D)** Loadings plot depicting the most variant features along the first principal component (corresponding to the x-axis for the scores plot in C). This figure can be generated using the data and code deposited in the "src" folder of https://github.com/dzhu8/Ad26-Dose-Down. Instructions to do so can be found in the README file. FcR, Fcγ receptor; NHP, nonhuman primate; NTD, N-terminal domain; PCA, principal component analysis; RBD, receptor-binding domain; SARS-CoV-2, Severe Acute Respiratory Syndrome Coronavirus 2; VP, viral particle.

the Holm–Bonferroni correction. Data suggest a more mature humoral immune response, consistent with a loss of IgM observed in the higher vaccine dosed animals (**Fig 3A**), and a significant enrichment of the Fc functional response that is corroborated by combining all titer, FcR binding, and functional features and comparing between protected and not protected animals (**S2 Fig**). Conversely, FcR binding features underwent the largest average fold change, depicted on the x-axis of the volcano plot, marked by >2-fold change and broad binding across nearly all, and especially, the phagocytic FCGR2A-3 receptor in protected animals.

Given the highly correlated nature of the vaccine-induced immune response, we next aimed to identify a minimal combination of features that best discriminated peak immune profiles in animals that were protected or infected after challenge. Least absolute shrinkage and selection operator (LASSO) regularization was used to reduce the degree of multicollinearity and identify a minimal set of immune features that tracked with protective immunity. As few as 7 features were sufficient to split the protected animals to those that became infected following challenge, as visualized by partial least squares discriminant analysis (PLS-DA) (**Fig 4D and 4E**). The protective features did not include neutralizing antibody levels, but instead included class switched IgG levels (IgG2, 4, and 3), ADCD, FcγR2A-3 binding, and ADCP activity, likely related to the presence of high neutralizing antibody levels across all animals. IgM was the sole feature associated with susceptibility, potentially marking an incompletely mature humoral immune response.

## The Fc effector profile is shaped by both RBD binding and non–RBD binding antibodies

To further dissect the relationship between neutralizing and Fc receptor binding antibodies, dose effects were examined across epitope-specific vaccine-induced antibodies. Given that the majority of neutralizing antibodies target the RBD [32], we probed the overall changes in RBD, NTD, and S2-specific titers and FcR binding across the dose-down groups (**Fig 5A**). A clear titer and FcR binding dose effect was observed across all 3 epitope-specific antibody classes, across both dominant neutralizing and nonneutralizing antibody targets. Differences were noted in some epitope specific populations, with a dampened NTD-specific FcγRIIa-3 response compared to NTD-specific binding to other Fc receptors, pointing to additional differences in epitope-specific evolution. Multivariate analysis of these epitope specific antibody responses demonstrated an enrichment of both RBD-targeting and S2-targeting features in protected animals with lower viral loads, supporting a protective role for both RBD-specific and non–RBD-specific antibody Fc effector functions (**Fig 5B**).

## Fc effector functions collaborate with neutralizing antibodies and T-cell immunity differently to achieve protection

Both antibodies and T cells have been linked to resolution of infection [18,22,32]. However, the precise contribution of neutralization, Fc effector function, and T-cell immunity remains

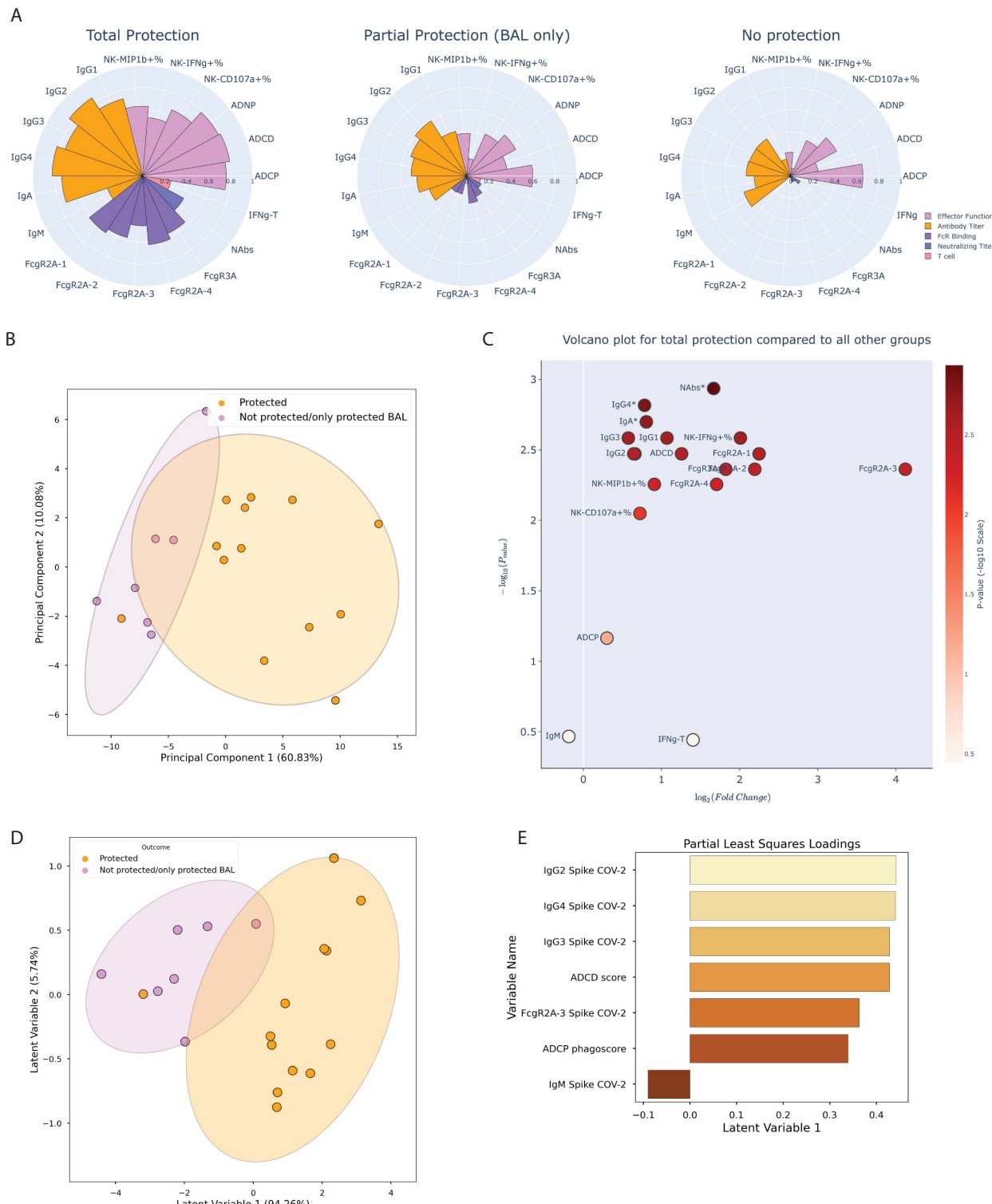

**Fig 4. Multivariate analysis of the immune response for varying degrees of protection against SARS-CoV-2 challenge identifies key determinants of immunity. (A)** Polar plots showing the mean percentile of each of the anti–SARS-CoV-2 antibody titers, FcR binding, and effector function activation features for NHPs, grouped by the degree of protection against viral challenge in the upper and lower respiratory tract; groups comprised NHPs protected in neither, one, or both compartments. Each wedge represents an individual feature, with colors representing the type of feature: orange: effector function; light purple: antibody titer; dark purple: FcR binding; blue: neutralizing antibody titer; and pink: T-cell IFNγ

secretion. **(B)** A latent variable scores plot resulting from PCA shows the degree of separation between the 4 vaccine doses that was achieved by analyzing anti–SARS-CoV-2 spike-component (anti-S1, S2, NTD, and RBD) features. Each point is an individual NHP, color coded by protection from infection (yellow: protected in BAL and nasal swab, purple: protected in BAL only or no protection in BAL or nasal swab). Ellipses illustrate the 95% confidence interval for dose groups. **(C)** A volcano plot showing the average fold change for each of the anti–SARS-CoV-2 spike features between completely protected NHPs and NHPs lacking complete protection, with the significance of the difference between these groups on the y-axis, quantified using the negative log *p*-value and measured using 2-sided Wilcoxon rank-sum tests. Each point represents an individual feature, color coded by the degree of statistical significance on a continuous colorbar. The star characters represent statistical significance following Holm–Bonferroni correction. Note that ADNP is not explicitly labeled due to overlap with IgG2, but its fold change is also significant. **(D)** A latent variable scores plot resulting from PLS-DA, with the variable regressed on being the categorical assignment of "protection" or "not protected/only protected in the lower respiratory tract," shows the degree of separation between NHPs that were completely protected and NHPs lacking complete protection that was achieved with anti–SARS-CoV-2 anti-Spike-component features. Each point is an individual NHP color coded by protection from infection (yellow: protected in BAL and nasal swab, purple: protected in BAL only or no protection in BAL or nasal swab). **(E)** Loadings plot depicting the most variant features along the first principal component (corresponding to the x-axis for the scores plot in D). This figure can be generated using the data and code deposited in the "src" folder of https://github.com/dzhu8/Ad26-Dose-Down. Instructions to do so can be found in the README file. ADNP, antibody-dependent neutrophil phagocytosis; BAL, bronchoalveolar lavage; FcR, Fcγ receptor; NHP, nonhuman primate; NTD, N-terminal domain; PCA, principal component analysis; PLS-DA, partial least squares discriminant analysis; RBD, receptor-binding domain; SARS-CoV-2, Severe Acute Respiratory Syndrome Coronavirus 2.

unclear. Thus, to explore the contribution of each "arm" of the immune system to protection, we created composite scores of our systems serology data using the partial least squares path modeling (PLS-PM) [33] framework to generate latent variable measurements (**Fig 6A**). Antibody features were divided into 3 groups including (1) antibody-mediated effector functional measurements (ADCP, ADNP, ADCC, NK-MIP1b+%, NK-CD107a+%, and NK-interferon gamma (IFNγ)+%); (2) binding antibody titers (IgG1, IgG2, IgG3, IgG4, IgA, and IgM); and (3) FcR binding measurements (FcγRA2-1, FcγRA2-2, FcγRA2-3, FcγRA2-4, and FcγRA3). The enrichment of these composite metrics as well as neutralizing antibody titers and T-cell immunity were compared across animals that remained uninfected or became infected after challenge. All 3 composite metrics and neutralizing antibody titers were significantly highly enriched among protected animals (**Fig 6B**). Thus, to define the degree of enrichment, a variable importance (VIP) coefficient was generated, comparing each variable to outcome as assessed by viral load. A clear hierarchy emerged across the 5 variables, with antibody titers representing the strongest correlate of protection, followed by antibody functions, FcR binding, neutralization, and T cells (**Fig 6C**). However, to gain a clearer sense of the precise role of each of the composite variables in providing protection from infection (blocking infection in both the upper and lower respiratory tract) or in controlling viral replication (viral levels after infection), VIP coefficients were generated for each outcome. The same hierarchy of features (**Fig 6C**) were associated with protection from infection (**Fig 6D**), with protection assessed using a binary variable indicating whether breakthrough was observed or not. Conversely, the arrangement of the 5 metrics changed with viral breakthrough (**Fig 6E**), when examining the association of each parameter with the level of viral replication only in the breakthrough animals. Specifically, antibody titers remained the top feature, followed by antibody functions, T-cell activity, FcR binding, and neutralizing antibodies were least critical for predicting viral control. These data highlight the importance of neutralization in the blockade of infection, T cells in the control of viremia, and antibody Fc effector functions as a critical collaborator and contributor to both prevention and control of viral infection.

## Antibody response to the SARS-CoV-2 mutant VOCs

As the coronavirus pandemic continues, new VOCs that are more infectious or escape the immune response are continuing to arise [34,35]. Current vaccines have been shown to be efficacious against the wild-type (WT) SARS-COV-2; however, the robustness of the variants to neutralization by either monoclonal or vaccine-induced binding antibodies [36–38] has prompted concerns. To address the impact of dose on emerging VOCs, we profiled the

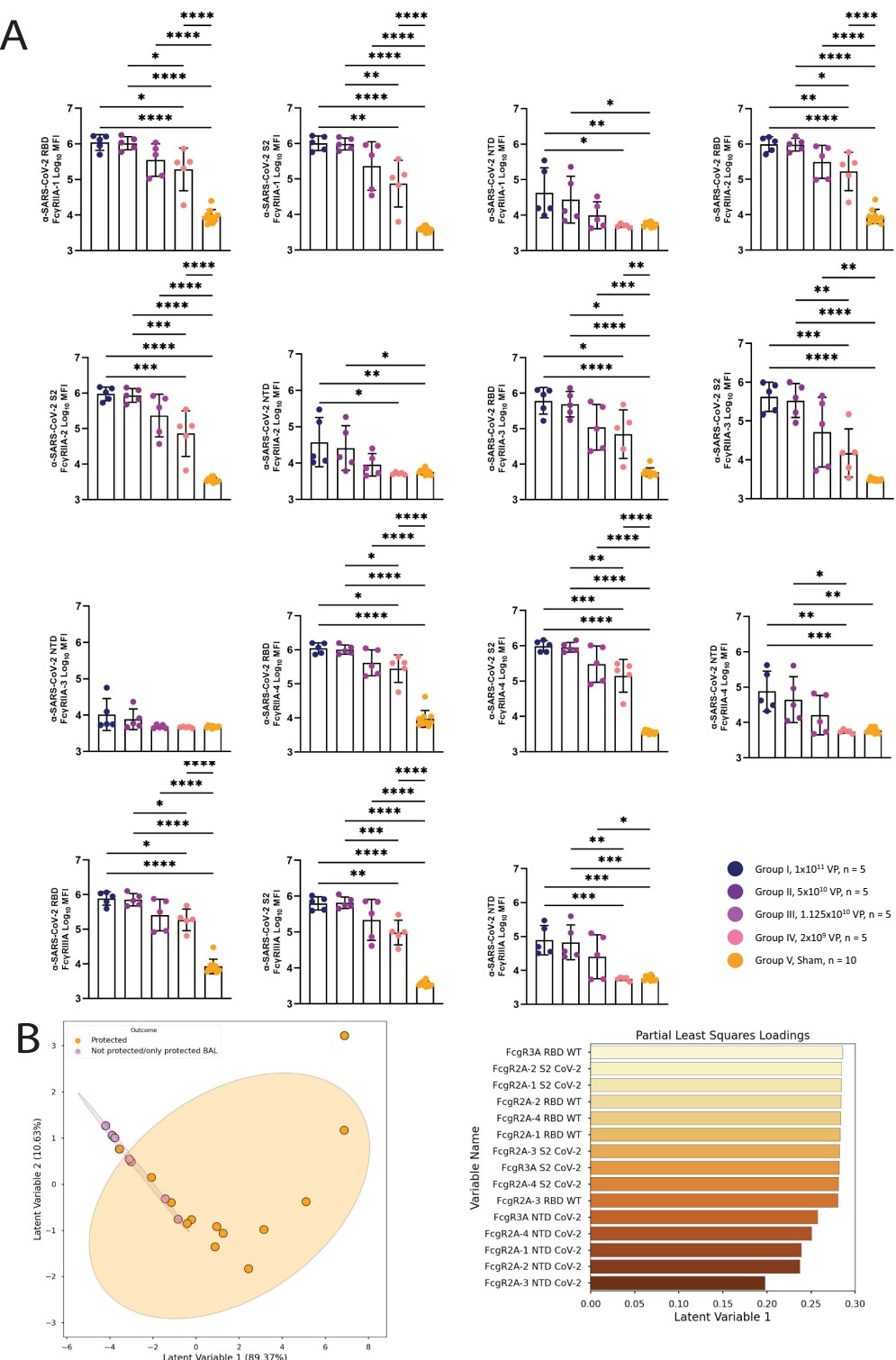

**Fig 5. Comparison of FcγR binding profiles across SARS-CoV-2 spike domains. (A)** The titer of anti–SARS-CoV-2 RBD, S2 spike subunit, and NTD spike subunit antibody binding to FcγRIIA-1, FcγRIIA-2, FcγRIIA-3, FcγRIIA-4, and FcγRIII were profiled using Luminex. **(B)** Scores plot (upper plot) and loadings along the first latent variable (lower plot) for a PLS-R model, regressing on the AUC of viral load as measured by BAL over a 10-day time course. Each point represents an NHP, either completely protected (orange) or lacking complete protection (purple); ellipses

correspond to the 95% confidence intervals for each group. Fig 5A data can be found in the file "data/figure5_data. csv." Fig 5B can be generated using the data and code deposited in the "src" folder of https://github.com/dzhu8/ Ad26-Dose-Down. Instructions to do so can be found in the README file. AUC, area under the curve; BAL, bronchoalveolar lavage; NHP, nonhuman primate; NTD, N-terminal domain; PLS-R, partial least square regression; RBD, receptor-binding domain; SARS-CoV-2, Severe Acute Respiratory Syndrome Coronavirus 2.

humoral response to multiple VOC antigens: D614G [39], N501YΔ69–70 (representing key mutations in Alpha/B.1.1.7 [40], and E484K (representing the key mutation in Beta/B.1.351 [41] and Gamma/P.1 [42], as well as being found in other variants of interest, such as Iota/ B.1.526 [43]). We observed relatively stable antibody binding across the variants in the highest doses with greater reductions to N501Yd69-70 or E484K in the lowest dose group, while FcR binding to the variants had a more profound loss across all doses (**Fig 7A**). This loss was linked to more profound loss of ADCP against the N501Yd69-70 and E484K variants, pointing to a potential weakness in the protection against VOCs in individuals with low vaccine-induced immunity. Composite analysis of the response to the VOCs highlighted robust titers and functionality in the top 2 dosed groups, but some loss of FcR binding and function in the lower dosed animals (**Fig 7B**). Comparisons of composite scores composed of the variant features showed significant differences between animals that were protected or infected after WT challenge, as well as high correlations between variant features and analogous WT features (**S3A– S3D Fig**). These data point to reduced Fc effector function against particular VOCs, linked to declining antibody titers, which may account for evidence of global breakthroughs, but in addition detectable levels of nearly all antibody effector functions across key mutations persisted that may contribute to global protection against severe disease in collaboration with T-cell immunity.

## Discussion

Despite the development of several efficacious vaccines against Coronavirus Disease 2019 (COVID-19) that provide protection against severe disease and death [5,7,15,44–46], understanding of the correlates of immunity have raised concerns about durability of protection against the multitude of emerging highly contagious VOCs. While neutralizing antibodies have emerged as a logical correlate of immunity to SARS-CoV-2, emerging signals of protection against VOCs-that evade neutralizing antibodies as well as in the setting of waning immunity point to the importance of alternate correlates of protection against SARS-CoV-2. The identification of the key immunologic mechanisms of protection could provide the critical insights to guide vaccine design and boosting regimens to provide durable protection to end this global pandemic.

While neutralizing antibody levels were identified as a strong correlate of protection in large scale vaccine efficacy trials, antibody binding titers were more robustly correlated with protection against severe disease across vaccine platforms [47,48]. Whether binding antibodies, with alternate nonneutralizing functions, represent direct mechanistic correlates of immunity or represent markers of a robust T-cell response remains unclear. However, both antibody-mediated Fc effector function and T-cell immunity have been proposed as markers of protective immunity [49–52]. Thus, to begin to define the role of neutralization, T-cell immunity, and alternate antibody mechanisms of action, we exploited a vaccine dose-down study in NHP to begin to tease apart the role of individuals arms of the immune response in protection against infection or disease. We found that in the context of an Ad26 vaccine, while antibody titers and neutralizing antibodies were robustly induced across most vaccine doses, T-cell responses were more sensitive to vaccine dose and were only detectable in animals

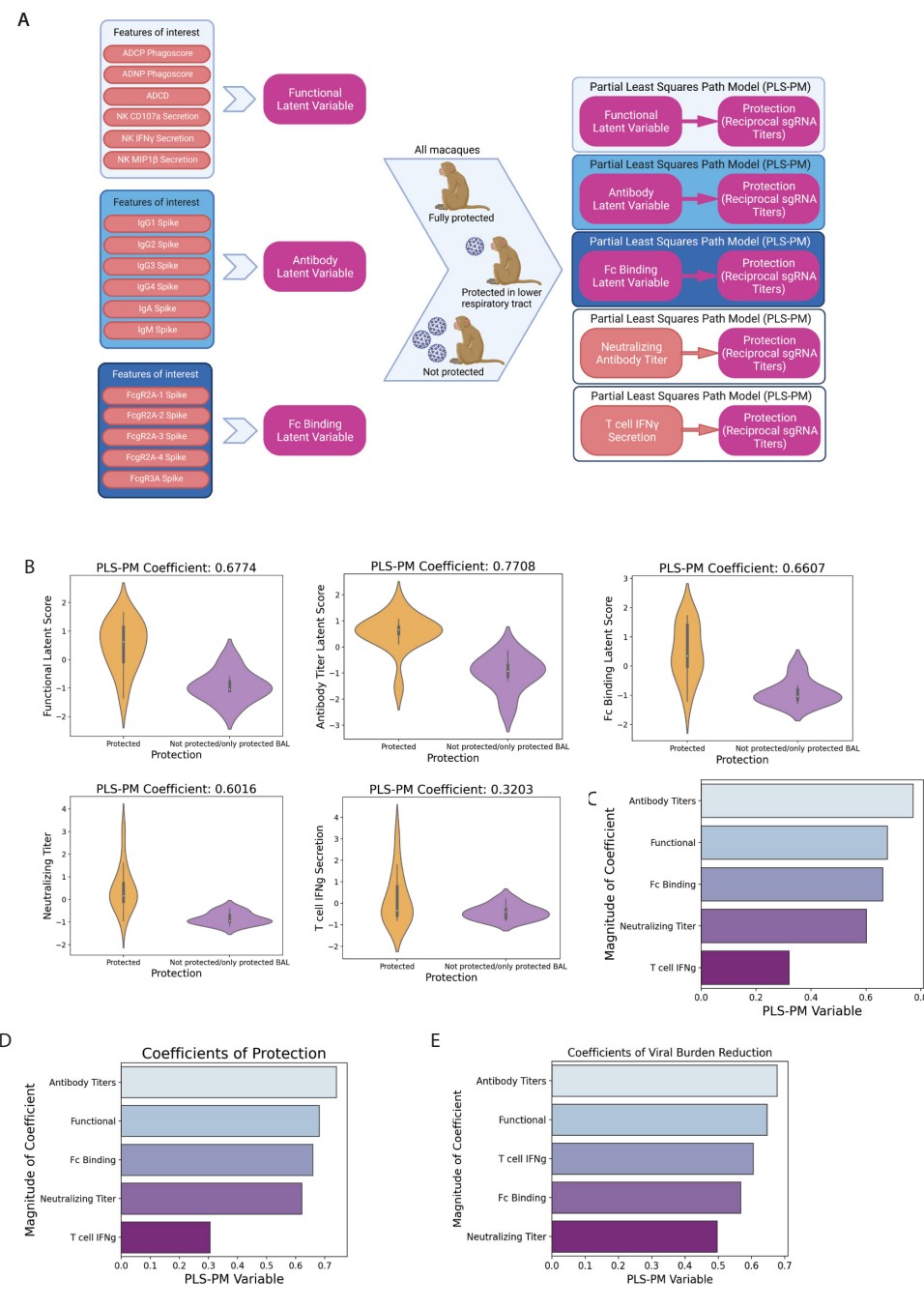

**Fig 6. Combined variable analysis identifies key combinatorial correlates of protection. (A)** Diagram depicting the process for computing latent variable coefficient scores with PLS-PM. Five PLS-PM models were computed, and the coefficients compared. Figure created with BioRender.com. **(B)** Violin plots showing the distribution of latent scores for the functional, antibody titer, and FcR binding features, as determined by PLS-PM, as well as for neutralizing antibody titer and T-cell IFNγ secretion. The white dot represents the median, the thicker bar the IQR, and the thinner bar 1.5 times the IQR in either direction. Thickness of the violin represents the density of a particular region. **(C)** Barplots showing coefficients for the PLS-PM analyses **(D, E)** Barplots showing coefficients for 2 PLS-PM analyses: with a binary protection variable ("fully protected" or "partially protected or not protected at all") (D) and with the protection variable being a combination of the AUC of cumulative viral titers measured at day 10 by both nasal swab and BAL (E). This figure can be generated using the data and code deposited in the "src" folder of https://github.com/dzhu8/Ad26-Dose-Down. Instructions to do so can be found in the README file. AUC, area under the curve; BAL, bronchoalveolar lavage; IFNγ, interferon gamma; IQR, interquartile range; PLS-PM, partial least squares path modeling.

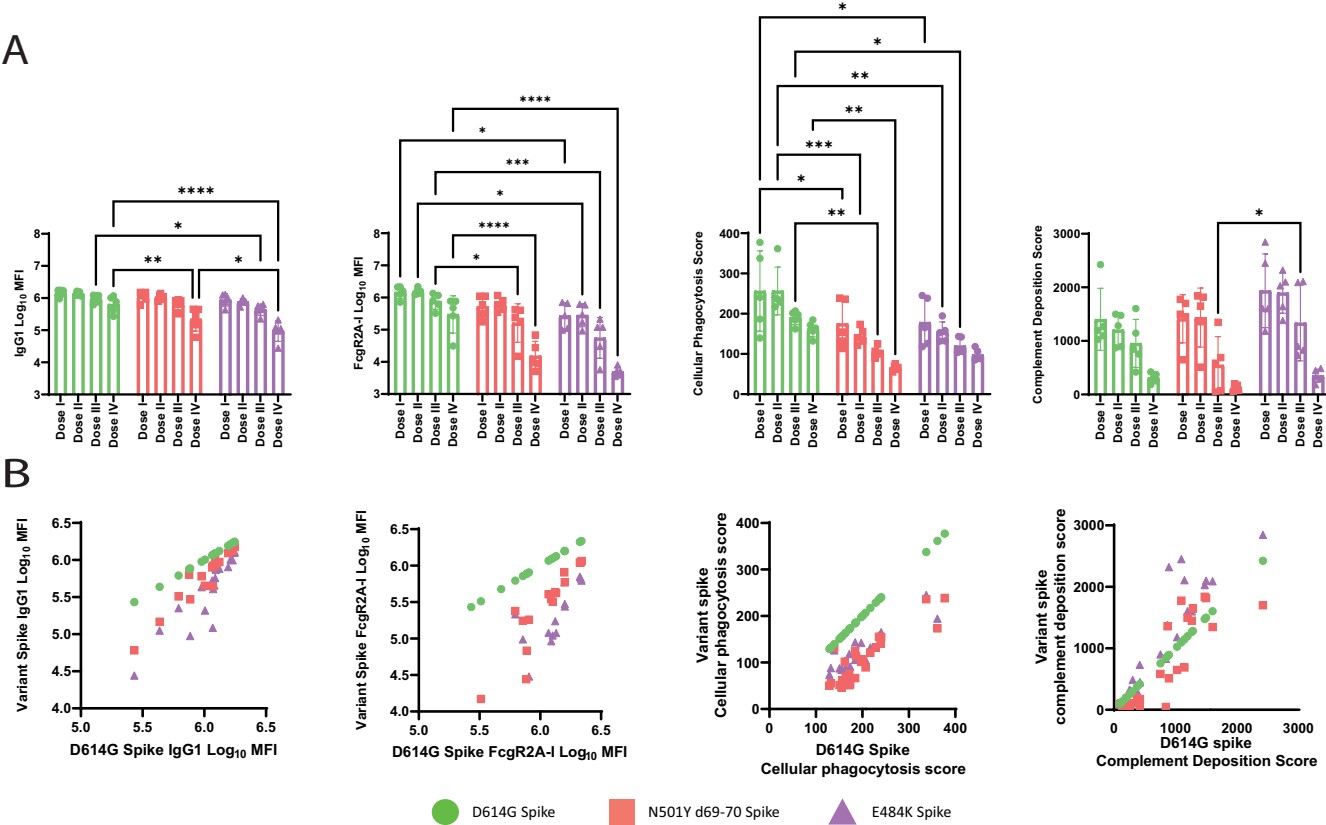

**Fig 7. Humoral response to VOCs. (A)** The humoral response of vaccinated NHPs was profiled against several SARS-CoV-2 spike proteins (Spike: D614G, N501YΔ69–70, and E484K). The antibody titers and FcR binding titers were assayed by Luminex and the functional responses were analyzed using their respective assays. The bars represent the median response and error bars represent standard deviation. **(B)** The dot plots show the correlation of the IgG1 titer, FcRgIIA-I binding titer, ADCP, or complement deposition between D614G SARS-CoV-2 and the variants N501YΔ69–70 and E484K. D614G is plotted as an example of perfect correlation. A 2-way ANOVA with Tukey correction for multiple comparisons was used to compare antibody levels between groups. Only significant comparisons are shown. ns, not significant, $^*p \leq 0.05$, $^{**}p \leq 0.01$, $^{***}p \leq 0.001$, $^{****}p \leq 0.0001$. This figure can be generated from the data found in data/figure7_data.csv of https://github.com/dzhu8/Ad26-Dose-Down. ADCP, antibody-dependent cellular phagocytosis; FcR, Fcγ receptor; MFI, median fluorescent intensity; NHP, nonhuman primate; SARS-CoV-2, Severe Acute Respiratory Syndrome Coronavirus 2; VOC, variant of concern.

immunized with doses equal to or greater than current clinically tested doses ($1 \times 10^{11}$ or $5 \times 10^{10}$ VPs). Conversely, antibody FcR binding and effector function were highly sensitive to vaccine dose, demonstrating a strong dose dependence, highlighting an unexpected disconnect in antibody titers/neutralization and T-cell and antibody functional induction. Whether these findings are generalizable to other vaccine formulations remains to be determined.

Breakthrough infections in the upper respiratory tract of animals immunized with the lower vaccine doses provided an opportunity to define immune correlates of protection. Because of the large number of variables captured using systems serology, the probability of an association with an Fc signal was larger. Thus, systems serology data were collapsed into 3 latent variables that captured antibody subclass/isotype diversification, FcR binding breadth, and Fc effector polyfunctionality. While this simplification of the data may obscure the precise correlate of immunity, this approach provides a more conservative means to compare additional antibody metrics to neutralization and T-cell immunity in the context of protective immunity. While T cells, neutralization, and Fc effector responses were all enriched in protected animals, the hierarchy of correlates of protection differed across outcomes. Specifically, antibody titers, FcR binding, antibody effector functions collaborated with neutralization in

the context of complete protection from infection (blockade of infection in the upper and lower respiratory tract). Conversely, antibody titers, FcR binding and antibody effector functions collaborated with T-cell immunity in the context of viral control in animals that experienced a breakthrough infection. These data support the notion that neutralization and Fc functions are key to blocking transmission, but that T cells collaborate with antibody functions to drive viral control and clearance.

The significant enrichment of several Fc effector functions in protected animals raises additional questions as to the particular Fc mechanism that may be most critical for protection from infection and/or disease. Among the Fc functions, NK cell activation was enriched in protected animals. However, early studies examining humoral correlates of immunity in natural COVID-19 infection pointed to a significant role for opsonophagocytic, but not NK cell activating antibodies, as correlates of resolution of severe disease [24]. Given the low abundance of NK cells in the upper respiratory tract prior to infection, it is unlikely that these cells contribute dominantly to the control/clearance of infection compared to the more highly abundant resident phagocytes and circulating granulocytes [53,54]. Moreover, because viral entry and release from target cells occurs via endocytosis and assembly on the Golgi (not the cell membrane), respectively, little to no Spike is likely expressed on the surface of infected cells [55]. Thus, in the absence of antibody-mediated recognition of Spike on infected cells, the role for NK cells in eliminating infected cells remains unclear. Conversely, Fc effector mediated capture and clearance of VPs may create a critical bottleneck, rapidly eliminating incoming or newly released viruses, providing a window for T cells to activate, proliferate, recognize, and eliminate infected cells, leading to the ultimate resolution of the infection. Thus, antibodies and T cells likely collaborate intimately to resolve SARS-CoV-2 infection, warranting further dissection to fully define the mechanistic correlates of immunity against SARS-CoV-2.

These data provide critical insights in the context of the current pandemic, where emerging VOCs have the capability of breaking through vaccine-mediated protection, but vaccine-induced immunity still elicits a response against several variants and provides some level of protection against severe disease and death [17,56–58]. Whether T cells or antibody effector function are more critical remains to be determined, but the analysis here points to a collaboration between vaccine-induced antibody Fc mechanisms and T cells, potentially via rapid antibody-mediated opsonophagocytic clearance and anamnestic boosting of T-cell immune responses that collectively lead to robust control and clearance of infection. While postimmunization samples could be linked to outcomes in this study, postchallenge humoral profiles will be important to compare prechallenge and postchallenge correlates of protection, and additionally to provide potential mechanistic links between the identified correlates of protection and containment of SARS-CoV-2.

The continued emergence of several SARS-CoV-2 variants with enhanced infectivity and immune evasive capacity have further complicated the worldwide vaccination effort [39,59–62]. The duration of the pandemic has been marked by decreased vaccine efficacy against these variants. Early trials for Novavax NVX-CoV2373 and Pfizer-BioNTech BNT162b2 saw a decrease from >90% efficacy against the WT to 50% to 60% against the Beta variant [15,63–65]. An exception was seen with the Delta variant, for which vaccines continue to confer >80% protection against severe disease and death [66]. Understanding vaccine-mediated correlates of protection and the contribution of Fc effector functions and T cells is critical in the context of estimating vaccine efficacy to new and emerging VOCs. Along these lines, despite the extraordinary global surge of the highly infectious Omicron variant, which significantly evades neutralizing antibodies [67–70], striking increases in infections were observed in the absence of a significant rise in severe disease and death. These data argue that while transmission occurred due to evasion of neutralizing antibody responses, additional vaccine-induced

mechanisms likely continue to contribute to the rapid control and clearance of the virus. Moreover, given the persistence of robust SARS-CoV-2 specific T cell [71–76] and Fc effector functions [77] across VOCs including Omicron, these arms of the immune response likely contribute to remains intact against the omicron strain of SARS-CoV-2. Thus, in line with the data explored here, which show a little drop in VOC binding, our results point to the potential mechanistic complementary importance of a collaboration between T cells and Fc-mediated effector functions, each playing a role in SARS-CoV-2 transmission and reduced disease. However, because our experiments used WA1/2020 antigens, it remains to be determined whether these observations will generalizable to other SARS-CoV-2 strains and vaccine inserts. Further experiments will need to be conducted to determine which Fc effector mechanism persists during Omicron infection.

The identification of correlates, and even surrogates of immunity, against the emerging VOCs is of the greatest urgency. While early studies against the original SARS-CoV-2 variant pointed to antibody titers and neutralization as key mechanisms of protection across vaccine platforms across the original SARS-CoV-2 variant, emerging data point to a central role of Fc effector functions and T-cell immunity as critical mechanistic players in antiviral control required for the prevention of severe disease and death. This study further demonstrates the key role of antibody effector functions and neutralization as mechanistic players in limiting transmission, but an alternate cooperation between Fc effector functions and T cells in limiting viral replication, a proxy of disease severity [78]. Thus, given the simplicity of measuring FcR binding antibodies, or simple opsonophagocytic functions, the latter utilized broadly for the licensure of bacterial vaccines, it is plausible that next generation assays aimed at measuring Fc biology across both existing and emerging VOCs could provide instrumental insights to guide next generation vaccine development or boosting.

## Materials and methods

### Cell line, viruses, and receptor

THP-1 cells (ATCC TIB-202) were maintained in Roswell Park Memorial Institute (RPMI) medium, supplemented with 10% FBS, 1% glutamine, 1% P/S, 1% HEPES, and 50 μM β-ME.

### Animal study design

The study design has been described [29]. In brief, 30 outbred Indian-origin adult male (10) and female (20) rhesus macaques (*Macaca mulatta*) were randomly allocated to groups. Animals received a single immunization of $1 \times 10^{11}$, $5 \times 10^{10}$, $1.125 \times 10^{10}$, or $2 \times 10^{9}$ vps Ad26.COV2.S (Janssen; $n = 5$/group) or sham ($n = 10$) by the intramuscular route without adjuvant at week 0. All animals were challenged with $1.0 \times 10^{5}$ TCID SARS-CoV-2, which was derived from USA-WA1/2020 (NR-52281; BEI Resources - Atlanta, Georgia, USA), by the intratracheal (IT) route at week 6. All serum samples were collected at week 6 postimmunization, before the challenge. All animals were housed at Bioqual (Rockville, Maryland, USA). Animal studies were conducted in compliance with all relevant local, state, and federal regulations and were approved by the Bioqual Institutional Animal Care and Use Committee (IACUC approval number 20-015P).

### Subgenomic mRNA assay

The assay has been described previously [29]. In brief, RNA was isolated from macaque BAL fluid and nasal swabs using the IndiSpin QIAcuba HT Pathogen Kit (Indical Bioscience - Leipzig, Germany). RNA was reverse transcribed to cDNA using Superscript VILO (Invitrogen - Carlsbad, California, USA) and stored at 4˚C until RT-PCR assays were performed. A Taqman

custom gene expression assay (Thermo Fisher Scientific - Waltham, Massachusetts, USA) was designed using the sequences target the E gene sgRNA [79]. The sequences for the custom assay were as follows, forward primer: sgLead-CoV2.FWD: CGATCTCTTGTAGATCTGTTCTC, E_Sarbeco_R: ATATTGCAGCAGTACGCACACA, E_Sarbeco_P1_(probe): VIC-ACACTAGC-CATCCTTACTGCGCTTCG-MGB. Reactions were performed on QuantStudio 6 and Flex Real-Time PCR systems (Applied Biosystems - Bedford, Massachusetts, USA). Analysis was performed on the QuantStudio Real-Time PCR Software (Life Technologies - Singapore). Standard curves were used to calculate sgRNA copies per mL or per swab.

## Anti-spike and RBD ELISA

The assay has been described previously [29]. In brief, 96-well plates were coated with 1ug/mL SARS-CoV-2 spike (S) or RBD protein in 1× DPBS and incubated at 4°C overnight. After incubation, plates were washed once with was buffer (0.05% Tween 20 in 1x DPBS) and blocked with 350-uL Casein block/well for 2 to 3 hours at room temperature. After incubation, block solution was discarded and plates were blotted dry. Serial dilutions of heat-inactivated serum diluted in casein block were added to wells and plates were incubated for 1 hour at room temperature, prior to 3 washes and a 1-hour incubation with a 1:1,000 dilution of anti-macaque IgG HRP (NIH NHP Reagent Program) at room temperature in the dark. Plates were washed 3 times, and 100 uL of SeraCare KPL TMB Stop solution per well. The absorbance at 450 nm was recorded using a VersaMax or Omega microplate reader.

## Pseudovirus neutralizing antibody assay

The assay has been described previously [29]. In brief, the packaging plasmid psPAX2 (AIDS Resource and Reagent Program), luciferase reporter plasmid pLenti-CMV Puro-Luc (Addgene), and spike protein expressing pcDNA3.1-SARS CoV-2 SDCT of variants were cotransfected into HEK293T cells by lipofectamine 2000 (Thermo Fisher Scientific). Pseudoviruses of SARS-CoV-2 variants were generated by using Wuhan prototype strain (Wuhan/WIV04/2019, GISAID accession ID: EPI_ISL_402124), D614G mutation, B.1.1.7 variant (GISAID accession ID: EPI_ISL_601443), or B.1.351 variant (GISAID accession ID: EPI_ISL_712096). The supernatants containing the pseudotype viruses were collected 48 hours posttransfection, which were purified by centrifugation and filtration with 0.45-mm filter. To determine the neutralization activity of the plasma or serum samples from participants, HEK293ThACE2 cells were seeded in 96-well tissue culture plates at a density of $1.75 \times 10^4$ cells/well overnight. Moreover, 3-fold serial dilutions of heat inactivated serum or plasma samples were prepared and mixed with 50 mL of pseudovirus. The mixture was incubated at 37°C for 1 hour before adding to HEK293T-hACE2 cells. Forty-eight hours after infection, cells were lysed in Steady-Glo Luciferase Assay (Promega - Madison, Wisconsin, USA) according to the manufacturer's instructions. SARS-CoV-2 neutralization titers were defined as the sample dilution at which a 50% reduction in relative light unit (RLU) was observed relative to the average of the virus control wells.

## IFNγ and IL-4 T cell ELISPOT assay

The assay has been described previously [29]. In brief, ELISpot plates were coated with mouse anti-human IFNγ monoclonal antibody (BD Pharmingen - San Diego, California, USA) or anti-human IL-4 monoclonal antibody (Mabtech - Stockholm, Sweden) at a concentration of 5 ug/well overnight at 4°C. Plates were washed with DPBS containing 0.25% Tween 20, and blocked with R10 media (RPMI with 11% FBS and 1.1% penicillin-streptomycin) for 1 hour at 37°C. The Spike 1 and Spike 2 peptide pools contain 15 amino acid peptides overlapping by 11 amino acids that span the protein sequence and reflect the N-terminal and carboxyl-terminal

halves of the protein, respectively. Spike 1 and Spike 2 peptide pools were prepared at a concentration of 2 mg/well, and 200,000 cells/well were added. The peptides and cells were incubated for 18 to 24 hours at 37°C. All steps following this incubation were performed at room temperature. The plates were washed with coulter buffer and incubated with Rabbit polyclonal anti-human IFN-g Biotin from U-Cytech (1 mg/mL), followed by a second wash and incubation with Streptavidin-alkaline phosphatase antibody from Southern Biotechnology (Birmingham, Alabama, USA) (1 mg/mL). The final wash was followed by the addition of Nitor-blue Tetrazolium Chloride/5-bromo-4-chloro-3-indolyl phosphate p-toludine salt (NBT/BCIP chromagen) substrate solution for 7 minutes (IFNγ) or 12 minutes (IL-4). The chromagen was discarded, and the plates were washed with water and dried in a dim place for 24 hours. Plates were scanned and counted on a Cellular Technologies Limited Immunospot Analyzer.

## ADCP and ADNP

ADCP and ADNP were conducted as previously described [80,81]. Briefly, spike or RBD proteins were biotinylated using EDC (Thermo Fisher Scientific) and EZ-link Sulfo-NHS-LC-LC (Thermo Fisher Scientific) and then coupled to yellow/green and then coupled to yellow/green FluoSphere NeutrAvidin-conjugated beads (Thermo Fisher Scientific, F8776). Immune complexes were formed by incubating the bead+protein conjugates with diluted serum (ADNP 1:50 dilution, ADCP 1:100 dilution) for 2 hours at 37°C and then washed to remove unbound antibody. The immune complexes were then incubated overnight with THP-1 cells (ADCP), or for 1 hour with RBC-lyzed whole blood (ADNP). THP-1 cells were then washed and fixed in 4% PFA, while the RBC-lyzed whole blood was washed, stained for CD66b Pacific Blue (BioLegend - San Diego, California, USA), CD3-AlexaFluro700 (BD Biosciences - Miami, Florida, USA), and CD14-APC-Cy7 (BD Biosciences) to identify neutrophils (CD3- CD14- CD66b+) and then fixed in 4% PFA. Flow cytometry was performed to identify the percentage of quantity of beads phagocytosed by THP-1 cells or neutrophils, and a phagocytosis score was calculated (% cells positive × median fluorescent intensity of positive cells). Flow cytometry was performed with an LSRII (BD), and analysis was performed using FlowJo V10.7.1.

## ADCD

ADCD was conducted as previously described [82]. Briefly, spike or RBD protein was biotinylated using EDC (Thermo Fisher Scientific) and EZ-link Sulfo-NHS-LC-LC (Thermo Fisher Scientific) and then coupled to red Neutravidin-conjugated microspheres (Thermo Fisher Scientific). Immune complexes were formed by incubating the bead+protein conjugates with diluted serum (ADCD 1:10 dilution) for 2 hours at 37°C and then washed to remove unbound antibody. The immune complexes were then incubated with lyophilized guinea pig complement (Cedarlane - Burlington, Ontario, Canada) and diluted in gelatin veronal buffer with calcium and magnesium (Boston BioProducts - Milford, Massachusetts, USA) for 30 minutes. C3 bound to immune complexes was detected by fluorescein-conjugated goat IgG fraction to guinea pig Complement C3 (MP Biomedicals - Irvine, California, USA). Flow cytometry was performed to identify the percentage of beads with bound C3 and a complement deposition score was calculated (% beads positive × median fluorescent intensity of positive beads). Flow cytometry was performed with an LSRII (BD) and analysis was performed using FlowJo V10.7.1.

## Antibody-dependent NK cell degranulation

Antibody-dependent NK cell degranulation was conducted as previously described [83,84]. Spike or RBD protein was coated on Maxisorp ELISA plate (Thermo Fisher Scientific) and then blocked with 5% BSA. Diluted serum (1:25 dilution) was then added and incubated for 2

hours at 37˚C. Human NK cells were isolated from peripheral blood by negative selection using the RosetteSep Human NK cell enrichment cocktail (STEMCELL Technologies - Vancouver, British Columbia, Canada) following the manufacturer's instructions. Human NK cells were then added to the bound antibody and incubated for 5 hours at 37˚C in the presence of RMPI+10% FBS, GolgiStop (BD), Brefeldin A (Sigma - Burlington, Massachusetts, USA), and anti-human CD107a-PE-Cy5 antibody (BD Biosciences). After incubation, cells were washed, stained with CD16-APC-Cy7, CD56-PE-Cy7, and CD3-Pacific Blue (BD Biosciences) and fixed in 4% PFA for 15 minutes. Intracellular staining was performed using the FIX/PERM Cell fixation and permeabilization kit (Thermo Fisher Scientific), and cells were stained for IFNγ-APC and macrophage inflammatory protein-1β-PE (BD Biosciences). Results were reported as percentage of NK cells positive for CD107a, IFNγ, or macrophage inflammatory protein-1β. Flow cytometry was performed with an LSRII (BD), and analysis was performed using FlowJo V10.7.1.

## Isotype and FcR-binding Luminex profiling

Isotyping and FcR profiling was conducted as previously described [84–86]. Briefly, spike, RBD, or variant proteins were carboxyl coupled to magnetic Luminex microplex carboxylated beads (Luminex - Austin, Texas, USA) using NHS-ester linkages with Sulfo-NHS and EDC (Thermo Fisher Scientific) and then incubated with serum (Isotypes 1:100 dilution, FcRs 1:1,000 dilution) for 2 hours at 37˚C. Isotyping was performed by incubating the immune complexes with secondary mouse-anti-rhesus antibody detectors for each isotype (IgG1, IgG2, IgG3, IgG4, and IgA) and then detected with tertiary anti-mouse-IgG antibodies conjugated to PE. FcR binding was quantified by incubating immune complexes with biotinylated FcRs (FcγR2A-1, FcγR2A-2, FcγR2A-III, FcγR2A-IV, and FcγR3A, courtesy of Duke Protein Production Facility) conjugated to Steptavidin-PE (Prozyme - Hayward, California, USA). Results were reported as median fluorescent intensity of PE staining for each antigen+bead combination. Flow cytometry was performed with an IQue (Intellicyt - Albuquerque, New Mexico, USA) and analysis was performed on IntelliCyt ForeCyt (v8.1).

## Multivariate analysis

All analyses were performed after removing the Sham samples and were performed using Python version 3.8.5. PCAs, PLS-DAs, and partial least square regressions (PLS-Rs) were performed using the "scikit-learn" package [87], with all measurements (antibody titers, FcR binding, ADCD, ADNP, ADCC, NK cell cytokine secretion, and degranulation) z-scored using scikit-learn's StandardScaler(). Samples were color coded on the PCA and PLS-DA visualizations using labels for either vaccine dose group or assignment of "protected" and "not protected/only protected BAL" based on the result of the viral load analysis. Ellipses were drawn for each of the labeled groups and extend 2 standard deviations in all directions based on the distribution of the corresponding group; they thus represent two-dimensional 95% confidence intervals. For the polar barplots, each feature is normalized by dividing by the maximum value of that feature within the dataset to obtain a percentile measurement, and the mean percentile measurement is computed for members of each group and represented on the plot. For the analyses described in **Fig 6**, PLS-PM [33] was performed using the "plspm" library in R version 4.1.0. Latent variables were comprised of a linear combination of all relevant WT spike-directed antibody features (with for example the "functional latent score" being a linear combination of ADCC, ADCP, ADNP, and NK cell degranulation and cytokine secretion). The value of a latent variable for any given sample was computed as a weighted mixture of these constituent features, with the weights being determined with the PLS-PM protocol, iterating

until convergence. The correlations heatmap was constructed using the "pandas" package [88]. Visualization for all multivariate analyses was accomplished using a combination of the "plotly" [89] and "matplotlib" [90] packages.

## Statistical analysis

Statistical analysis for the results described in **Fig 3** onward was performed using Python 3.8.5. In the analyses described in **Fig 6**, measurements of antibody, FcR binding, and functional enrichments as well as neutralizing antibody and T-cell IFNγ secretion were represented using violin plots. The region enclosed by each box represents the interquartile range of the variable for a particular group, and the line bisecting each box is the median. For the analysis represented in **S3B Fig**, each functional enrichment score was calculated by taking an average of the Z-score for each measured variant spike-directed antibody feature (with the exception of B.1.1.7 and B.1.351, for which RBD-directed antibody features were used; for example, "functional enrichment" is the average z-score for ADCC, ADCP, NK cell degranulation, etc.). Mann–Whitney U tests were used to determine statistical significance of differences between groups for both the violin plots and the volcano plot, as normality could not be assumed for these features. Similar analysis was performed to generate **S2 Fig**, using WT spike-directed features. Two different Python packages were used to conduct these tests, "statannot" and "scipy" [91]. A *p*-value of $\leq 0.05$ was considered significant for an individual test, with this *p*-value adjusted for multiple hypothesis testing using Holm–Bonferroni corrections (% beads positive × median fluorescent intensity of positive beads). Flow cytometry was performed with an LSRII (BD), and analysis was performed using FlowJo V10.7.1.

## Supporting information

**S1 Fig. Correlation between each anti–SARS-CoV-2 spike feature, neutralizing antibodies, and T-cell function, between each vaccine dose group. (A–D)** Correlation plots showing the $r^2$ correlation between each immune feature among all vaccine regimens. (A) Group 1 $1 \times 10^{11}$ VPs, (B) Group II, $5 \times 10^{10}$ VPs, (C) $1.125 \times 10^{10}$ VPs, and (D) $2 \times 10^9$ VPs. This figure can be generated using the data and code deposited in the "src" folder of https://github.com/dzhu8/Ad26-Dose-Down. Instructions to do so can be found in the README file. SARS-CoV-2, Severe Acute Respiratory Syndrome Coronavirus 2; VP, viral particle.
(EPS)

**S2 Fig. Enrichment scores for WT spike-directed features. (A)** The violin plots show the distribution of each labeled feature between completely protected NHPs (orange) and NHPs lacking complete protection (purple). Functional, antibody, and Fc enrichment were calculated by computing the average z-score of all WT spike-directed effector functions, antibody titers and Fc receptor binding measurements, respectively. This figure can be generated using the data and code deposited in the "src" folder of https://github.com/dzhu8/Ad26-Dose-Down. Instructions to do so can be found in the README file. NHP, nonhuman primate; WT, wild-type.
(EPS)

**S3 Fig. Further profiling of the humoral response to VOCs. (A)** The humoral response was profiled against several additional variants and SARS-CoV-2 proteins (Spike: WT, D614G, N501Y, E484K, B.1.1.7, K417N; RBD: WT, N501Y, E484K, B.1.351, P.1). The squares represents the z-scored value for each immune feature (x-axis) for each NHP (y-axis). Z-score was calculated across each individual column. **(B)** The violin plots show the distribution of each labeled feature between completely protected NHPs (orange) and NHPs lacking complete protection (purple). Functional, antibody and Fc enrichment were calculated by computing the

average z-score of all anti-variant effector functions, antibody titers and Fc receptor binding measurements, respectively. **(C)** Scatterplots of WT spike directed enrichment scores against variant-directed enrichment scores, with the computed Spearman correlation included in the plot title. Each point represents an NHP, either completely protected (orange) or lacking complete protection (purple). **(D)** The PCA scores plot and associated loadings show the degree of discrimination achieved using only the combination of functional, antibody and Fc enrichment scores created with anti-variant features. Each point represents an NHP, either completely protected (orange) or lacking complete protection (purple); ellipses correspond to the 95% confidence intervals for each group. This figure can be generated using the data deposited in the "data" folder and code deposited in the "src" folder of https://github.com/dzhu8/Ad26-Dose-Down. Instructions to do so can be found in the README file. Data for S3A Fig can be found in "data/figure7_data." NHP, nonhuman primate; PCA, principal component analysis; SARS-CoV-2, Severe Acute Respiratory Syndrome Coronavirus 2; VOC, variant of concern; WT, wild-type.
(EPS)

## Author Contributions

**Conceptualization:** Matthew J. Gorman, Dan H. Barouch, Galit Alter.

**Data curation:** Daniel Y. Zhu, Matthew J. Gorman, Dan H. Barouch.

**Formal analysis:** Daniel Y. Zhu, Matthew J. Gorman, Douglas A. Lauffenburger.

**Investigation:** Matthew J. Gorman, Dansu Yuan, Jingyou Yu, Noe B. Mercado, Katherine McMahan, Erica N. Borducchi, Michelle Lifton, Jinyan Liu, Felix Nampanya, Shivani Patel, Lauren Peter, Lisa H. Tostanoski.

**Resources:** Laurent Pessaint, Alex Van Ry, Brad Finneyfrock, Jason Velasco, Elyse Teow, Renita Brown, Anthony Cook, Hanne Andersen, Mark G. Lewis.

**Software:** Daniel Y. Zhu.

**Supervision:** Douglas A. Lauffenburger, Galit Alter.

**Validation:** Daniel Y. Zhu, Matthew J. Gorman.

**Visualization:** Daniel Y. Zhu, Matthew J. Gorman, Galit Alter.

**Writing – original draft:** Daniel Y. Zhu, Matthew J. Gorman, Galit Alter.

**Writing – review & editing:** Daniel Y. Zhu, Matthew J. Gorman, Douglas A. Lauffenburger, Dan H. Barouch, Galit Alter.

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
