## [Editor Report · Decision Letter 0]

13 Oct 2021

Dear Galit, 

Thank you for submitting your manuscript entitled "Defining the determinants of protection against SARS-CoV2 infection and viral control in a dose down Ad26.CoV2.S vaccine study in non-human primates" for consideration as a Research Article by PLOS Biology.

Your manuscript has now been evaluated by the PLOS Biology editorial staff and I am writing to let you know that we would like to send your submission out for external peer review.

Once your full submission is complete, your paper will undergo a series of checks in preparation for peer review. Once your manuscript has passed the checks it will be sent out for review. 

If your manuscript has been previously reviewed at another journal, PLOS Biology is willing to work with those reviews in order to avoid re-starting the process. Submission of the previous reviews is entirely optional and our ability to use them effectively will depend on the willingness of the previous journal to confirm the content of the reports and share the reviewer identities. Please note that we reserve the right to invite additional reviewers if we consider that additional/independent reviewers are needed, although we aim to avoid this as far as possible. In our experience, working with previous reviews does save time. 

If you would like to send your previous reviewer reports to us, please specify this in the cover letter, mentioning the name of the previous journal and the manuscript ID the study was given, and include a point-by-point response to reviewers that details how you have or plan to address the reviewers' concerns. Please contact me at the email that can be found below my signature if you have questions. 

Please re-submit your manuscript within two working days, i.e. by Oct 15 2021 11:59PM.

Kind regards,

Paula

Paula Jauregui, PhD

Associate Editor

PLOS Biology

---

## [Decision Letter · Decision Letter 1]

9 Dec 2021

Dear Dr. Alter,

Thank you for submitting your manuscript "Defining the determinants of protection against SARS-CoV2 infection and viral control in a dose down Ad26.CoV2.S vaccine study in non-human primates" for consideration as a Research Article at PLOS Biology. Your manuscript has been evaluated by the PLOS Biology editors, an Academic Editor with relevant expertise, and by several independent reviewers.

In light of the reviews (below), we will not be able to accept the current version of the manuscript, but we would welcome re-submission of a much-revised version that takes into account the reviewers' comments. We cannot make any decision about publication until we have seen the revised manuscript and your response to the reviewers' comments. Your revised manuscript is also likely to be sent for further evaluation by the reviewers.

In particular, we think it is important to show whether nAb or non-nAb are responsible for the Fc-mediated effects. We also think that the experiments suggested by reviewer #2 would certainly be interesting and valuable, but we understand that working with live SARS-CoV-2 requires special permissions and this might not be possible in a restricted time frame, in which case these experiments won't be required for publication. Please, address the rest of the reviewers concerns. 

We expect to receive your revised manuscript within 3 months. 

**IMPORTANT - SUBMITTING YOUR REVISION**

*Re-submission Checklist*

*Published Peer Review*

*PLOS Data Policy*

*Blot and Gel Data Policy*

Sincerely,

Paula

---

Paula Jauregui, PhD

Associate Editor

PLOS Biology

REVIEWS:

Reviewer #1: Immune response against virus.

Reviewer #2: Antibody response against virus.

Reviewer #3: Immunologist.

Reviewer #1: This is beautiful study that attempts to discern the correlates of protection for SARS-CoV-2 vaccination. Through detailed analyses the authors link neutralizing antibodies to protection from infection and T-cell responses to control of viremia in infected animals. They provide evidence that Fc-dependent antibody functions complement neutralizing antibody responses and T-cell responses to confer protection and viral control, respectively. I have minor questions/comments for the authors.

- The authors should make an attempt to address where the Fc functions are coming from. Are these functions mediated by the neutralizing antibodies, or are they originating from non-neutralizing antibodies? Is there any way to absorb neutralizing antibodies and assess the importance of the Fc functions mediated by non-neutralizing antibodies? This would be very informative regarding the value of binding non-neutralizing antibodies for protective outcomes against SARS-CoV-2 exposure and infection.

- The authors did not complete the query about where data may be found. Currently, this box contains "XXX".

- Line 101 in the legend for Figure 1 - "relates" should be "relate".

- For the data in Figure 2, the authors make the claim that the lower two doses of the vaccine induce significantly lower responses. However, as presented in the Figure, significance is only apparent for the lowest dose for most variables. The text should be amended to more closely reflect the data as presented.

- Legend for Figure 3 - the colors listed do not correspond to the colors used in the actual figure. 

- Figure 6 - no statistical analyses are shown.

Reviewer #2: While vaccine-induced neutralizing antibody responses represent the major correlate of protection against infection and severe disease, there are other antibody-mediated responses that can contribute to protection aside from neutralization. Here, Gorman and colleagues sought to uncover other mechanisms by which vaccines provide protection by dosing down a currently adenoviral platform, Ad26.CoV.2.S, that is the basis of the Johnson and Johnson COVID-19 vaccine. By dosing down the vaccine, they hypothesize they can reveal other important arms of the immune response that are involved in protection, which may often be saturated by current dosing regimens. They found that neutralizing and binding antibody titers were minimally impacted by dosing. However, Fc-effector functions and T cell-mediated immunity were significantly more impacted in the dosed-down group. The authors claim that having robust Fc- and T cell-mediated responses are crucial for controlling viral replication. Overall, the data presented are of sufficient quality in rigor and generally supports the authors' claims. However, the method by which Fc-mediated responses are measured may not truly reflect what may occur during natural infection. This should be addressed by the authors as it has broad implications on how the data concerning the activation of Fc-mediated is interpreted and used as a correlate of protection. There is also some information that may have been unintentionally left out that could further clarify the data for readers. These will be further discussed below.

Strengths:

The experimental design of dosing down a vaccine dose in non-human primates is an excellent approach in revealing arms of the immune response that may or may not be impacted by dosing. One can then correlate specific immunological functions that may be sufficient or deficient with disease phenotype.

Weaknesses:

Based on Figure 1, samples were collected post immunization and challenge (at week 6). However it is unclear if the data presented in Figures 2 to 6 are from which timepoints - are they serum titers after the immunization and at which time point, or are they after the challenge? It was not clearly indicated in the main text or in the figure legend. Please clarify. Also, after clarification, is there a good correlation in the magnitude of the immune response post immunization to post challenge in a dose-dependent manner? This should be further discussed in the text or discussion. 

SARS-CoV-2 buds through the ER-Golgi pathway. In fact, the major subcellular localization of the S in the presence of other viral proteins (i.e., M) during natural infection accumulates in the early Golgi (PMID: 25855243, 17166901, 15507643). It is noted that others have used cells stably expressing only the S as target cells, however, it has been demonstrated that the expression of S alone allows for surface expression (PMID: 34504087). It is also noted that a large number of studies have used FluoSphere NeutrAvidin or similar technology use it to couple the spike of a viral protein as a surrogate for phagocytosis assays. While the data presented is of high rigor, it is not known whether the S is sufficiently expressed on the surface of infected cells in nature. At most it may represent how antibodies encounter viral particles, form immune complexes and are taken up by phagocytes through some Fc-mediated mechanism. However, due to how the present authors and others would like to highlight the importance of Fc-mediated immunity in controlling viral replication, one would have to use infected cells as target cells and look for activation of Fc-dependent activation of innate immune cells as an accurate measure. The thinking behind this is that clearance of infected cells requires the target of antibodies be expressed on the surface of infected cells. It would increase confidence in the interpretation of the data if the authors can confirm that indeed the S is the target of Fc-mediated immune response in naturally infected cells - as a mechanism of clearing virally-infected cells. 

Reviewer #3: The manuscript submitted by Alter and colleagues uses an innovative dose-down vaccine and challenge model as an approach to define correlates of Ad26.CoV2.S vaccine-mediated protection against SARS-CoV2 in the rhesus macaque model. This work is critically important. Despite success of current vaccine modalities, a clear understanding of correlates of protection remains to be defined. In the current study, the authors demonstrate that neutralizing, binding, and FcR-dependent functional antibody responses were involved in protection, while antibody binding, FcR-dependent functional responses and T cell responses were involved in viral control. Thus, FcR-dependent antibody responses contribute to both protection and control. This suggests a need to evaluate these types of antibody responses against existing and emerging variants to guide vaccine boosting and next-generation vaccine development. The manuscript is well written, well designed, and expected to have broad and substantial impact. There are only a few minor concerns to address.

* Lines 105-106 what is meant by "systems serology was applied to the aforementioned BAL and NS samples"? From the methods it would seem that all antibody assays were performed using serum. Please confirm this is the case and clarify in the methods/results. Also, all immune measurements were performed at Week 6 correct? This should also be clearly stated in the methods and the week 2 and 4 collection timepoints should be removed from Figure 1 if they are not relevant to the current study.

* For ELISPOT assay, were the 200,000 cells/well whole PBMC or T cell enriched?

* Speculation in discussion on lines 342 and 343 should be removed. The T cell data included does not differentiate between CD8 and CD4, and TFH have not been profiled. Too many assumptions are involved in this statement.

* Can font size on polar plots be increased?

* In the discussion, please reiterate that the results and conclusions are limited to the Ad26-based vaccine. These findings cannot yet be generalized to the other vaccine platforms.

---

## [Editor Report · Decision Letter 2]

18 Mar 2022

Dear Dr Alter,

On behalf of my colleagues and the Academic Editor, Sarah Rowland-Jones, I am pleased to say that we can in principle accept your Research Article "Defining the determinants of protection against SARS-CoV2 infection and viral control in a dose down Ad26.CoV2.S vaccine study in non-human primates" for publication in PLOS Biology, provided you address any remaining formatting and reporting issues. These will be detailed in an email that will follow this letter and that you will usually receive within 2-3 business days, during which time no action is required from you. Please note that we will not be able to formally accept your manuscript and schedule it for publication until you have completed any requested changes.

IMPORTANT: Many thanks for providing a Github link to your underlying data. I've asked my colleagues to also request that you include this link in all relevant main and supplementary Figure legends. 

PRESS

Sincerely, 

Dario

Dario Ummarino, PhD 

Senior Editor 

PLOS Biology

dummarino@plos.org